# Abnormalities in Regional Cerebral Blood Flow Due to Headache in a COVID-19 Infected Patient Observed on 99mTC-ECD Brain SPECT/CT

**DOI:** 10.3390/reports6040058

**Published:** 2023-12-04

**Authors:** Ya-Chun Chu, Shin-Tsu Chang, Hung-Yen Chan, Daniel Hueng-Yuan Shen, Hung-Pin Chan

**Affiliations:** 1Department of Nuclear Medicine, Kaohsiung Veterans General Hospital, Kaohsiung City 81314, Taiwan; kristin920920@gmail.com (Y.-C.C.); hongyenchan0407@yahoo.com.tw (H.-Y.C.); hyshen@vghks.gov.tw (D.H.-Y.S.); 2Department of Physical Medicine and Rehabilitation, Kaohsiung Veterans General Hospital, Kaohsiung City 813414, Taiwan; ccdivlaser1959@gmail.com; 3Department of Physical Medicine and Rehabilitation, Tri-Service General Hospital and National Defense Medical Center, Taipei City 11490, Taiwan; 4Institute of Biomedical Sciences, College of Medicine, National Sun Yat-sen University, Kaohsiung City 804, Taiwan

**Keywords:** 99mTc-ECD brain SPECT/CT, COVID-19 infection, headache, pain-matrix, regional cerebral blood flow

## Abstract

A 31-year-old man was diagnosed with a COVID-19 infection, presenting with a diffuse headache and an uneven sensation of fullness, despite having no prior systemic diseases. 99mTc-ECD Brain SPECT/CT revealed a marked decrease in blood perfusion in the cerebral cortices, predominantly in the frontal region and involving the olfactory sulcus. In contrast, there was increased perfusion in the occipital lobe and asymmetrical perfusion in the cerebellum. This case highlights changes in regional blood flow perfusion that might affect the functional connectivity of the pain matrix, leading to the onset of headaches and associated underlying mechanisms.

SARS-CoV-2, which began as a coronavirus outbreak in late 2019, is a highly transmissible disease that led to a global pandemic characterized by acute respiratory illnesses and posed significant threats to public health. While the majority of COVID-19 patients primarily exhibit respiratory tract symptoms, there is a growing body of literature documenting neurological symptoms. These have been investigated using functional imaging techniques like brain SPECT/CT and positron emission tomography/computerized tomography (PET/CT) [1]. Osawa et al. documented a case where a patient developed central nervous system syndrome post-COVID-19 infection, exhibiting symptoms of myoclonus and ataxia. A subsequent brain N-isopropyl-p-(123I)-iodoamphetamine (IMP) SPECT/CT revealed hypoperfusion in the frontal cortex [2]. They concluded that brain perfusion SPECT is effective in identifying functional changes in patients infected with COVID-19. In another study, a brain 18F-FDG PET/CT scan of a COVID-19 patient showcased cognitive impairment linked with frontal FDG hypometabolism [3]. It has been reported that the virus may increased the bioelectrical activity of the brain [4]. The authors posited that the virus might penetrate the central nervous system via the olfactory bulb, potentially extending to other areas such as the limbic thalamus, cerebellum, or brainstem [5,6,7]. However, there are limited reports on regional brain perfusion associated solely with headaches, devoid of other severe neurological symptoms. Headache causes may be linked to alterations in the brain’s pain processing pathway, commonly referred to as components of the pain matrix [8,9]. The spinothalamic system is considered the primary pathway for transmitting nociceptive and thermoceptive information to the cerebral cortex. The spinothalamocortical pathway relays interoceptive information to central structures, which then project to the viscerosensory cortex in the mid-insula, as well as the right anterior insula and orbitofrontal cortices [10]. Instances of facial pain, accompanied by reduced blood flow in the prefrontal cortex, suggest that alterations in blood flow might underpin pain due to a lack of inhibition in cortical and limbic structures [11]. Honda et al. documented a decrease in rCBF in the prefrontal area, right orbitofrontal cortex, and anterior cingulate gyrus using SPECT in patients with chronic pain [12]. Changes in brain blood flow perfusion might be pivotal in functional connectivity alterations within the pain matrix. This is particularly evident in the orbitofrontal cortices as observed in our patient. The alterations is in rCBF, especially within the brain’s pain matrix extending from the olfactory bulb to other regions, might be mechanisms causing headaches in this COVID-19 infected patient. We hypothesize that cytokine release, in response to post-infectious antibody- or cell-mediated immune activation, may cross the blood-brain barrier and induce central inflammatory responses. These responses are thought to be associated with low cerebral blood flow, as indicated in previous studies [13,14]. The frontal lobe, particularly the prefrontal cortex, is involved in mediating antinociceptive effects and pain processing. This is achieved through its connections to other areas of the cerebral neocortex, hippocampus, periaqueductal gray, thalamus, amygdala, and basal nuclei [15]. Lower blood perfusion in the frontal lobe, induced by a central inflammatory response, may decrease the threshold for nociception, leading to frequent headache attacks in our case. In well-equipped medical facilities, regular use of SPECT/CT studies in COVID-19 infected patients with neurological symptoms could be part of routine examination, especially in patients with severe symptoms. Though more evidence is required to correlate brain perfusion abnormality with clinical symptoms, more clues of the invasion route of COVID-19 virus can be gathered, which could further prompt treatment targeting the mechanism of the fatal virus. The following case highlights changes in regional blood flow perfusion that might affect the functional connectivity of the pain matrix, leading to the onset of headaches and associated underlying mechanisms (Figure 1).

## Figures and Tables

**Figure 1 reports-06-00058-f001:**
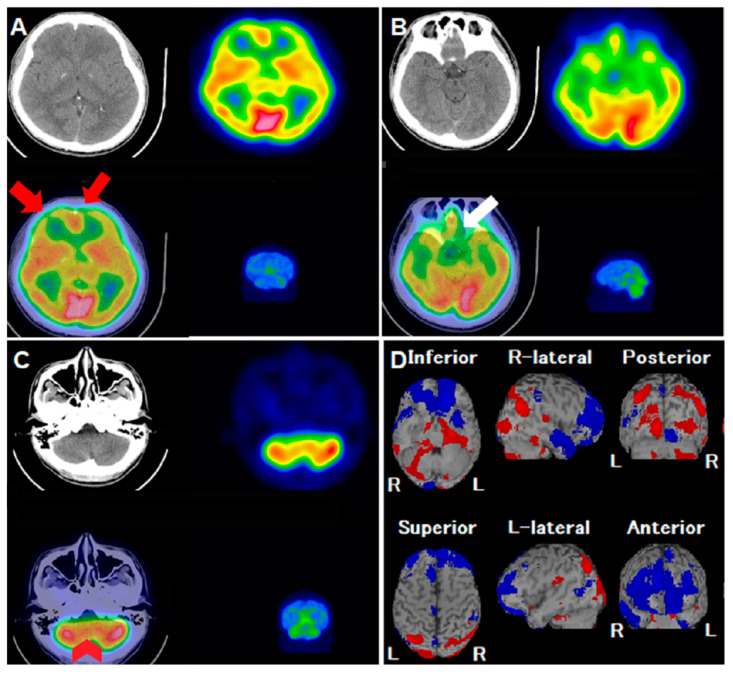
A 31-year-old man experienced intermittent headaches and lower back pain, particularly during work. Despite the progression of his symptoms, a conventional CT scan did not show any abnormalities. To assess possible neurological or psychological causes, his physician recommended a 99mTc-ethyl cysteinate dimer (ECD) single-photon emission computerized tomography/computerized tomography (SPECT/CT). However, after undergoing the SPECT/CT examination, the patient developed severe rhinorrhea, sore throat and general malaise several hours later. He was subsequently diagnosed with COVID-19 (SARS-CoV-2) and consistently reported a diffuse headache along with an uneven sensation of fullness in his head, though he had no visual, sensory, or balance disturbances. The patient reported chronic tension headache for long time. However, during COVID infection period, headache was more severe than he ever experienced. Pain was accompanied with band-like tightness and diffuse fullness sensation, without nausea, vomiting, pulsatile sensation, photophobia or phonophobia. There was no autonomic symptoms including lacrimation, miosis, or redness of the eye. Symptoms lasted for about 1 week. After recovery from COVID-19 infection, there was no residual neurological deficit. A review of the 99mTc-ECD SPECT images revealed hypoperfusion defects in the bilateral frontal lobes ((**A**), axial views; red arrows) and the olfactory cortex ((**B**), axial views; white arrow). There was also asymmetrical perfusion activity in both cerebellum hemispheres ((**C**), axial views; red arrowheads), with no corresponding structural abnormalities. An additional analysis using the Easy Z-score imaging system (eZIS) highlighted a significant reduction in regional cerebral blood flow (rCBF), prominently in the bilateral middle/inferior frontal gyri and bilateral olfactory sulci ((**D**), blue-colored areas), contrasted by hyperperfusion in the occipital lobe. This case highlights the regional brain perfusion abnormalities identified by 99mTc-ECD SPECT/CT in a patient infected with COVID-19. Despite presenting with headaches, the patient showed no structural abnormalities, underscoring the potential neurological implications of the disease.

## Data Availability

The data underlying this study are available in this article.

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
