# Peer review of "Abnormalities in Regional Cerebral Blood Flow Due to Headache in a COVID-19 Infected Patient Observed on 99mTC-ECD Brain SPECT/CT"

_reports, 2023, doi:10.3390/reports6040058_

Round 1

Reviewer 1 Report

Comments and Suggestions for Authors

Dear Authors,

Thank you for giving me the opportunity to review your article.

It deals with an interesting case of SARS-CoV-2 infection associated with neurological symptoms. The value of high-definition CNS imaging techniques is highlighted in your article.

I suggest you detail the time- interval between the moment of the SPECT/ CT investigation and the onset of respiratory symptoms related to COVID-19 in this patient. Please also consider to briefly describe the clinical course and the neurological outcome of this patient. 

Would you recommend the regular use of SPECT/ CT studies in COVID-19 patients with neurological symptoms?     

Author Response

Thank you for comments. Please see the attachment for our reply. 

Sincerely

Dr Chan

Reviewer 2 Report

Comments and Suggestions for Authors

The paper presented to me for review deals with the interesting and new issue of the effect of COVID-19 infection on abnormalities in brain regional cerebral blood flow in a patient with headaches. 

However novel the paper is, presenting evidence of direct effects of SARS-CoV2 virus on brain structures - which we observe clinically - it requires several additions before being considered for publication.

1. the key issue is to complete what type of headache the patient was diagnosed with and what was the history of the headache - did he suffer from any type of headache - for example, migraine - before becoming ill? what were the features of the current headache - besides being "diffuse headache and an uneven sensation of fullness" was it accompanied by nausea, vomiting, hypersensitivity to light, sounds? any additional neurological symptoms? was this the first such headache in his life?

2. the discussion is worth supplementing with similar cases confirming metabolic disturbances in the brain during COVID-19 infection based on: PMID: 34119843

Author Response

(The authors gave the same response as above.)

Reviewer 3 Report

Comments and Suggestions for Authors

The title of your study suggests a causal relationship between decreased regional blood flow and the incidence of headaches. However, upon reviewing the manuscript, it appears that the co-occurrence of reduced blood flow and headaches in your observations might be coincidental rather than causative. There is a lack of concrete evidence in the study that directly links the decrease in regional blood flow to the onset of headaches. Consider discussing alternative explanations for your findings and providing a more nuanced view of the relationship between blood flow changes and headaches.

Author Response

Dear reviewer:

Thank you for your comments. Please see the attached file for our response. 

Sincerely

Dr Chan
